# SRUM: Fine-Grained Self-Rewarding for Unified Multimodal Models

## Abstract

Recently, remarkable progress has been made in Unified Multimodal Models (UMMs), which integrate generation and understanding capabilities within a single framework. However, a key challenge remains: a model's powerful understanding often fails to transfer into complex image generation. This often occurs because the understanding and generation modules are trained separately or leading an internal conflict during co-training. As a result, a model can accurately assess a prompt against an image but cannot generate a correct image from that same prompt. To resolve this challenge, we introduce SRUM, the self-rewarding post-training framework designed to improve the model to align its generation with its understanding module. Without needing any new human-labeled data, SRUM creates a self-improvement loop where the model's own understanding module acts as an internal "evaluator", providing corrective feedback by rewarding to its generation module. Our core innovation is a two-part reward system that offers comprehensive guidance: comprising a **global reward** for overall compositional structure and a **local reward** for fine-grained, object-level fidelity. This multi-scale feedback proves critical for complex generation. SRUM sets a new state of the art and strong generaliztion, boosting performance as on T2I-CompBench from 82.18 to **88.37** and on T2I-ReasonBench from 40.7 to **50.4** in image accuracy. Overall, our work establishes a powerful new paradigm for enabling the UMMs' understanding module to guide its own generation.

## 1 Introduction

Text-to-Image (T2I) models have achieved remarkable progress in generating high-quality and diverse images from given prompts (Ramesh et al., 2021; Saharia et al., 2022; Podell et al., 2024). However, they often fail to accurately interpret instructions involving world knowledge, complex spatial relationships, detailed attribute binding, or compositional reasoning (Huang et al., 2023). These limitations point to a fundamental lack of deep semantic understanding of T2I models.

To address this challenge, researchers have developed Unified Multimodal Models (UMMs) based on large multimodal models. UMMs represent a promising direction by integrating both understanding and generation capabilities within a single framework (Wu et al., 2024b;a; Dong et al., 2024; Xie et al., 2024). By sharing a common backbone for the two core capabilities of multimodal understanding and generation, UMMs possess the inherent potential for synergy, offering a path to resolve the comprehension challenges that are difficult for standard T2I models.

A key challenge for UMMs is that their training methods often fail to unlock the full potential of their advanced architecture. For simplicity, the most common strategy is to train the understanding and generation modules separately (Tong et al., 2024a; Chen et al., 2025b; Pan et al., 2025). While practical, this approach creates a disconnect, preventing the model's understanding capabilities from being effectively transferred to its generation module. Alternatively, some models jointly train two modules but the gradient conflicts between different tasks make them unable to promote each other (Xie et al., 2025b; Wang et al., 2024d). This leads a significant capability gap: the model's understanding module consistently outperforms its generation capabilities. Model can often confirm if an image matches a prompt but can't generate the image from the text alone Figure 1. Consequently, the key to unlocking the full potential of UMMs lies in bridging this internal gap. The challenge is to harness the model's innate understanding to guide and improve its generative process (Zhou et al.,

2024). Targeting this pivotal challenge, we introduce a novel self-rewarding method during the post-training stage called **S**elf-**R**ewarding for **U**nified Multimodal **M**odels **(SRUM)**.

Our core insight is that the solution to this internal conflicts lies within the UMMs' architecture itself. The model's **generation module** can act as the "generator", while its powerful **understanding module** with function of grounding and judging can serve as the internal "evaluator". This establishes a natural, closed-loop system for self-rewarding without scoring by external judgment model. However, a simple, holistic score is insufficient for complex compositional tasks. As our ablation studies later confirm, such coarse feedback fails to provide the nuanced guidance required for meaningful improvement. Therefore, we introduce a fine-grained judgment and scoring framework that decomposes the internal reward into two synergistic components. First, to ensure the overall scene structure aligns with the prompt, we introduce a **global reward** to assess compositional coherence. Second, to enforce precise, object-level fidelity, we employ a **local reward** that provides fine-grained feedback on specific image regions, addressing attribute binding and semantic accuracy.

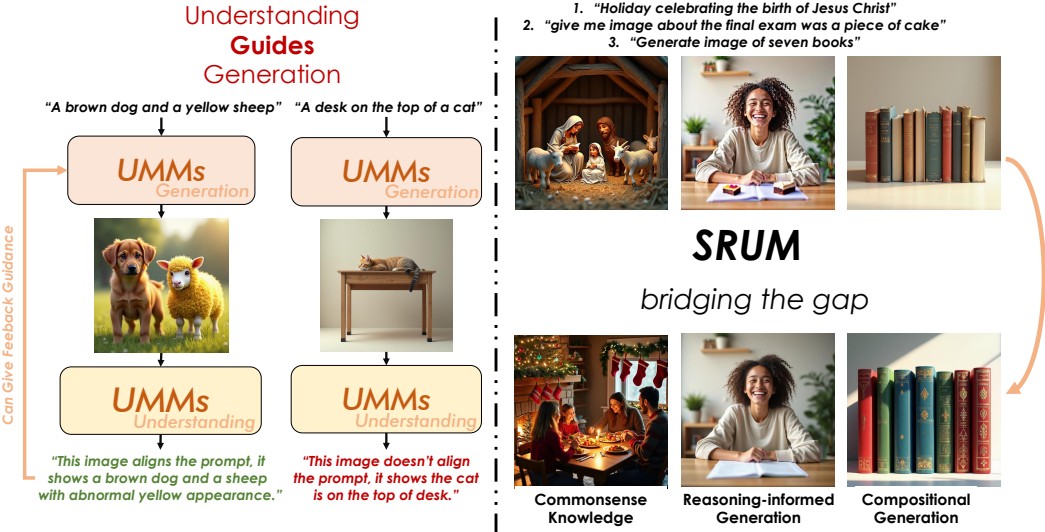

Figure 1: The example on the left indicates that the capability of the current UMMs' understanding module surpasses that of its generation module: the understanding module can reasonably identify mismatches between the generated content and the prompt, whereas the generation module is prone to producing incorrect candidates based on the given prompt in relevant cases. This not only highlights a gap between understanding and generation but also reveals the potential for understanding to guide generation. Inspired by this insight, we propose **SRUM** to bridge this gap, particularly in complex generation domains..

Through extensive experiments, we demonstrate that our approach significantly improves the composition, reasoning, and visual fidelity of UMM, and demonstrates generalization across in-domain and out-of-domain settings. SRUM achieves SOTA results on the T2I-CompBench and T2I-ReasonBench, improving the overall score of a strong baseline model from 82.18 to 88.37 in composition and from 40.7 to 50.4 in image accuracy with given prompts. Our key contributions can be summarized as follows:

1. We are the first to propose and implement a more mature self-rewarding framework for UMMs during post-training stage, successfully bridging the gap between their advanced understanding and generation modules through a self-improvement loop.

2. We introduce a novel decomposed reward design that combines global compositional assessment with local object-level feedback, providing multi-scale and fine -grained guidance that our ablations show is critical for performance.

3. We not only achieve superior performance on complex compositional benchmarks but also demonstrate strong generalization to in-domain and out-of-domain tasks. Ultimately, SRUM provides a powerful paradigm for the under- standing module to guide the generation module of UMMs.

## 2 RELATED WORKS

### 2.1 ARCHITECTURES FOR UNIFIED MULTIMODAL MODELS

Unified Multimodal Models (UMMs) have emerged as a prominent research direction, aiming to integrate diverse tasks like visual understanding and generation within a single, end-to-end trained architecture. Recent architectural paradigms can be broadly categorized. The **Purely Autoregressive (AR)** approach extends the next-token prediction paradigm of LLMs to visual data, treating images as a sequence of discrete tokens (Team, 2024; Wang et al., 2024d). A key refinement in this area involves decoupling the visual encoders, using a semantic encoder for understanding tasks while retaining a reconstruction-based tokenizer for generation (Wang et al., 2024e; Team et al., 2025), as demonstrated by Janus (Wu et al., 2024a). Show-O further refines this by integrating a discrete-diffusion schedule to improve token prediction (Xie et al., 2024). More prevalent are hybrid architectures that combine the strengths of AR and diffusion models. One major category consists of **Sequential AR-Diffusion** models, where an AR component generates an intermediate representation that conditions a diffusion-based decoder. In some variants, a pre-trained MLLMs is kept frozen for reasoning, and its features are routed via learnable queries or hidden states to an external image generator (Tong et al., 2024a; Shi et al., 2024; Lin et al., 2025). This cascaded design effectively leverages powerful existing models. A more integrated approach uses a **Unified Transformer Backbone** (Zhao et al., 2024; Chen et al., 2024a), where both AR and diffusion objectives are optimized simultaneously within a single transformer. To improve scalability, the **Mixture-of-Transformers (MoT)** paradigm has been introduced (Liang et al., 2025; Deng et al., 2025). This approach, exemplified by Bagel, employs a sparse, modular design where specialized experts handle different modalities but share information through a common attention mechanism.

### 2.2 POST-TRAINING STAGE IN UMMS

In addition to architectural innovations, considerable research has focused on post-training strategies to enhance the generative abilities of UMMs. Methods such as Chain-of-Thought (CoT) and test-time verification introduce explicit reasoning steps or iterative output validation (Guo et al., 2025b; Fang et al., 2025; Duan et al., 2025). However, these often depend on external models and do not fundamentally improve the native generative capacity of the UMMs. Reinforcement learning techniques—including Direct Preference Optimization (DPO) and Group Relative Policy Optimization (GRPO)—leverage human or automated feedback to refine generation policies. While effective, these require carefully curated paired data and delicate advantage function tuning with text dependent rewards (Rafailov et al., 2023; Guo et al., 2025a). Reconstruction Alignment (RecA) introduces a post-training method based on reconstruction loss, yielding improved semantic understanding (Xie et al., 2025a). Some work has also attempted to use rule-level rewards for guidance, but this is not universal and needs to be designed for different tasks (Hong et al., 2025; Mao et al., 2025). In contrast, SRUM operates without additional data generation. It leverages the model's inherent understanding to score self-generated samples and incorporates them into training, thereby enhancing performance.

### 2.3 SELF-REWARDING IN UNDERSTANDING MODELS

Self-rewarding mechanisms have emerged as a significant paradigm for enhancing the understanding and reasoning capabilities of MLLMs. These approaches aim to reduce reliance on external preference data by enabling models to generate their own reward signals, thereby facilitating continuous self-improvement. For instance, CSR (Zhou et al., 2024) achieves zero-cost self-enhancement through iterative online DPO with visual constraint rewards. SRPO (Choi et al., 2024) introduces a two-stage reflective reward mechanism, significantly improving the quality of reflection and answer accuracy in complex reasoning tasks. R1-Reward leverages process consistency rewards and stable reinforcement learning algorithms to enhance long-range reasoning stability (Guo et al., 2025a). Collectively, these works signal a paradigm shift from external rewards to self-criticism and optimization. Our SRUM framework proposes a more holistic approach.

## 3 CONSTRUCTION OF SELF-REWARDING UNIFIED MULTIMODAL MODELS STEP BY STEP

This section details the pipeline of our SRUM. Our process begins with the generation of high-quality image candidates and their corresponding bounding boxes using a Unified Multimodal Models (UMMs) (Section 3.1). These candidates are then meticulously evaluated by a dual-level prompts that assesses both local fidelity and global composition. Subsequently, the scores from this evaluation are transformed into a dense, spatially-aware reward map during the rewarding process (Section 3.2). Finally, this reward map is integrated into a novel reward-weighted training, which allows for targeted, region-specific model refinement while preventing reward hacking (Section 3.3).

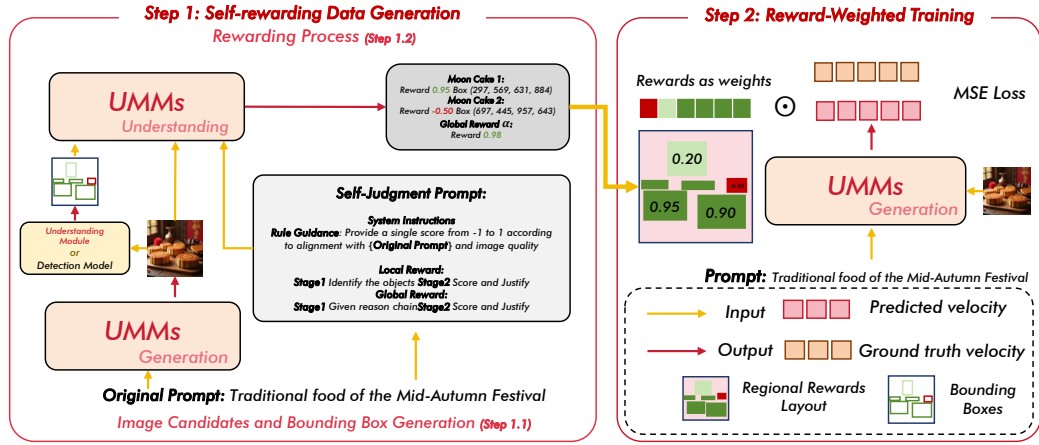

Figure 2: Showcase of the pipeline of the **SRUM**. Including the rewards generation steps, the design of regional rewards, and how to apply them to the generation end for training.

### 3.1 IMAGE CANDIDATES AND BOUNDING BOX GENERATION

As depicted in Figure 2, our pipeline begins by synthesizing a set of candidate images using a Unified Multimodal Models (UMMs) conditioned on input prompts. To ensure high-fidelity outputs, this generative process leverages the "think" mode or called CoT mode of the Bagel (Deng et al., 2025). Subsequently, we produce bounding box proposals for each image using either the UMMs' internal understanding module or a lightweight external detector like SAM (Kirillov et al., 2023). Finally, to enable precise grounding and reward modeling, the understanding module filters these proposals, retaining only those semantically aligned with the initial prompt.

### 3.2 REWARDING PROCESS

**Self-Judgment Prompt Design**. Then, we devise a dual-level judgment mechanism to assess image quality and prompt alignment, building upon recent work in automated evaluation (Xu et al., 2023; Zhang et al., 2023b; Lin et al., 2024; Ghosh et al., 2023). Our approach first performs a local judgment of object fidelity and artifacts using a strict $[-1.0, 1.0]$ scoring scale, where a mandatory "Reason" field elicits an interpretable rationale akin to chain-of-thought prompting (Guo et al., 2025b; Fang et al., 2025). We enforce semantic grounding by ensuring identified objects correspond to prompt keywords, and a non-linear penalty maps severe distortions to a high-penalty negative range (e.g., -0.9 to -0.5) to reflect human visual sensitivity. Subsequently, a global judgment evaluates the holistic composition and spatial alignment with the prompt's intent. Crucially, for prompts lacking specific compositional directives (e.g., "a picture of a tree"), a neutral score range (e.g., -0.4 to 0.4) is applied. This avoids unfairly penalizing plausible layouts when no specific arrangement was requested, thereby ensuring a solid assessment.

**Rewarding Process**. Next, we leverage the UMMs' inherent grounding capabilities to generate fine-grained reward scores for all relevant image regions, including both foreground objects and background, that are relevant to the given prompt. Our scoring mechanism consists of both a local and a global reward. To ensure meaningful aggregation, the global reward is normalized to the $[0, 1]$ range. This prevents the product of two negative values from yielding a spurious positive reward

signal (see Appendix Section D for details). All regional rewards are ultimately aggregated into a dense reward map, enabling its integration into our training.

## 3.3 REWARD-WEIGHTED TRAINING

The core of our reward-weighted training is the reward-driven term $\mathcal{L}_r$, which operates on the model's velocity prediction $v_\theta$ from a standard practice in flow-based diffusion frameworks (Liu et al., 2023b; Lipman et al., 2023; Esser et al., 2024). This term is modulated by two distinct feedback signals: a regional reward map $R \in [-1, 1]$ for localized refining, and a global scalar $\alpha$ that assesses overall compositional quality, provided by an understanding module. The product of these signals, $\alpha \cdot R$, weights the squared error between the predicted velocity $v_\theta$ and the target velocity derived from the original latent $x_0^{\text{gt}}$. This mechanism enables fine-grained control, encouraging preservation where feedback is positive ($\alpha \cdot R > 0$) and repulsion where it is negative ($\alpha \cdot R < 0$). This use of rewards to guide the training objective is inspired by preference optimization techniques (Rafailov et al., 2023):

$$\mathcal{L}_r = \mathbb{E}\left[\alpha \cdot R \odot \left(v_\theta - (\epsilon - x_0^{\text{gt}})\right)^2\right] \tag{1}$$

Second, to ensure that the output of the model conforms to the desired overall structure and prevents reward hacking, we introduce a constraint term. This term acts as a regularizer by penalizing the squared $\ell_2$ distance to the target velocity of the artifact-free and $x_0^{\text{gt}}$:

$$\mathcal{L}_{\text{ref}} = \mathbb{E}\left[\left\|v_\theta - (\epsilon - x_0^{\text{gt}})\right\|^2\right] \tag{2}$$

The final training objective is a weighted sum of these two losses, balanced by a tunable hyperparameter, $\lambda_c$:

$$\mathcal{L}_{\text{Total}} = \mathcal{L}_r + \lambda_c \cdot \mathcal{L}_{\text{ref}} \tag{3}$$

This composite design enables targeted local refinement while maintaining global coherence. It also inherently prevents reward hacking, thereby safeguarding the generated output distribution against significant distortion.

## 4 ANALYSIS OF SELF-REWARDING: GENERALIZATION AND PRINCIPLES

We validate our Self-Rewarding for Unified Multimodal Models (SRUM) method across various unified multimodal models (UMMs) and evaluation benchmarks. In particular, we investigate the following aspects:

- **Generality and Performance:** SRUM achieves state-of-the-art (SOTA) performance on complex compositional text-to-image generation benchmark and delivers consistent performance gains across different UMM frameworks, demonstrating its broad applicability. (Table 1)

- **Component Efficacy:** Ablation studies confirm that each component of the SRUM framework makes a critical contribution to the overall performance. (Figure 3)

- **Generalization:** SRUM demonstrates robust in-domain and out-of-domain generalization, indicating that its improvements stem from enhanced reasoning capabilities rather than data memorization. (Tables 3 to 5)

### 4.1 EXPERIMENTAL SETUP

**Model Architectures.** We evaluate SRUM on two powerful open-source UMMs. All experiments are conducted as a post-training phase, starting from the official pre-trained weights. **Bagel** (Deng et al., 2025) is a versatile UMM that serves as our primary model for comprehensive analysis, including main results, ablation studies, and generalization tests. We evaluate both its standard and Chain-of-Thought (CoT) inference modes. **Blip3o** (Chen et al., 2025a) is another state-of-the-art UMM used to validate the generality and effectiveness of our proposed SRUM method.

**Datasets and Benchmarks.** Our experiments leverage several specialized datasets for training and evaluation to ensure a thorough and multi-faceted analysis. For consistent and objective scoring across all generation benchmarks, we employ QwenVL-2.5-72B (Bai et al., 2025) as the designated multimodal evaluator. Our experiment begins with instruction data sourced from the

T2I-CompBench training set (Huang et al., 2023). For our primary evaluation, we use the standard split of the same benchmark to compare SRUM-enhanced models against leading T2I and UMMs' baselines. To assess generalization, we evaluate the model's in-domain transferability on GenEval (Ghosh et al., 2023) and WISE (Niu et al., 2025), which feature similar compositional challenges, without any fine-tuning. Furthermore, we test broader, out-of-domain reasoning capabilities on T2I-ReasonBench (Sun et al., 2025), a benchmark containing complex prompts that require knowledge beyond the training distribution.

## 4.2 MAIN RESULTS

Our main results are presented in Table 1, which compares leading Text-to-Image (T2I) models and Unified Multimodal Models (UMMs) on the T2I-CompBench standard split. To ensure a stable and consistent assessment, we employed the QwenVL-2.5-72B (Bai et al., 2025) model as the multimodal evaluator for evaluating the results, not for our rewarding algorithm.

The results clearly show that incorporating our method, SRUM, yields substantial and consistent performance gains across nearly all compositional categories. Notably, Bagel$_{+SRUM}$ with Chain-of-Thought (CoT) achieves the highest overall score among UMMs at 88.37. This represents a significant 3.91-point improvement over its base CoT version and a 6.19-point gain over the standard Bagel model, affirming SRUM's ability to enhance these architectures.

A detailed breakdown reveals that SRUM's impact is most pronounced in categories requiring sophisticated structural and logical reasoning. For instance, BLIP3o$_{+SRUM}$ sets a new SOTA score of 93.88 in the Spatial category, demonstrating superior handling of object positioning. Similarly, Bagel$_{+SRUM}$ with CoT reaches a new peak of 88.60 in 3D Spatial, indicating an improved grasp of complex layouts. These gains extend to other challenging tasks, with both Bagel$_{+SRUM}$ variants showing marked improvement in Numeracy.

However, despite the global performance boost, we observed a nuanced trade-off. For example, while BLIP3o$_{+SRUM}$ excels in structural tasks, it exhibits a slight performance decrease in the Texture category compared to its baseline.

Table 1: **Comprehensive T2I-CompBench Results.** This table includes T2I (Labs, 2024; Esser et al., 2024; Podell et al., 2024) and Unified Multimodal Models (Chen et al., 2025b; Xie et al., 2025b). Models incorporating the SRUM are denoted with a subscript. **Bold values** indicate the highest score in each respective column under. Green values indicate the improvements.

| Model | 3d spatial | Color | Complex | Nonspatial | Numeracy | Shape | Spatial | Texture | Overall |
|---|---|---|---|---|---|---|---|---|---|
| *T2I Models* | | | | | | | | | |
| FLUX.1-dev | 76.39 | 90.63 | 83.51 | **87.47** | **75.30** | 80.20 | 84.23 | 87.07 | 83.10 |
| FLUX.1-schnell | **79.38** | 84.53 | 81.96 | 85.55 | 72.82 | 82.20 | 85.49 | 86.38 | 82.29 |
| SD-3-medium | 77.83 | **91.63** | **84.73** | 86.12 | 72.80 | **83.72** | 88.20 | **89.03** | **84.26** |
| SD-xl-base-1 | 72.25 | 77.75 | 75.00 | 85.28 | 57.14 | 72.18 | 77.08 | 78.38 | 74.38 |
| *Unified Multimodal Models* | | | | | | | | | |
| Janus-Pro | 76.17 | 84.25 | 80.28 | 80.47 | 56.43 | 65.14 | 79.67 | 69.67 | 74.01 |
| Show-o2 | **88.61** | 87.73 | 87.88 | 85.91 | 69.74 | 73.99 | 86.60 | 82.17 | 82.83 |
| OmniGen2 | 82.21 | 92.22 | 86.87 | 88.51 | 72.00 | 83.95 | 90.07 | **90.88** | 85.84 |
| BLIP3o | 81.73 | 89.92 | 85.55 | 84.78 | 71.67 | 83.75 | 92.47 | 87.45 | 84.66 |
| Bagel | 77.98 | 89.30 | 83.32 | 85.03 | 70.40 | 81.94 | 81.52 | 87.93 | 82.18 |
| Bagel (CoT) | 84.66 | 88.85 | 86.10 | 85.64 | 75.36 | 84.33 | 82.71 | 88.07 | 84.46 |
| BLIP3o$_{+SRUM}$ | 83.78 | 90.22 | 86.57 | 85.10 | 74.52 | **85.44** | 93.88 | 86.52 | 85.75 |
| Bagel$_{+SRUM}$ | 83.10 | **92.90** | 88.69 | 88.47 | 78.52 | 84.23 | 86.92 | 89.57 | 86.55 |
| Bagel$_{+SRUM}$ (CoT) | 88.60 | **92.90** | **91.31** | 90.48 | **80.12** | 84.47 | 89.93 | 89.15 | **88.37** |

## 4.3 EMPIRICAL STUDY

We primarily employed three basic models for Bagel analysis: **Base Model,** Bagel's open-source weights are used directly for inference. **SFT Model,** Bagel generates images based on training instructions, then directly trains the model itself to create a self-training SFT model. **SRUM Model,**

Bagel generates images according to the training instructions, and then uses the SRUM training to obtain the final evaluation model.

**Ablation Results**. To further verify the effectiveness of our proposed reward configuration, we perform an ablation study on the results of Bagel on T2I-CompBench by systematically modifying the reward scheme. As shown in the Figure 3, we experimented with several variants, including a sample-level reward, a binarized reward, the removal of the KL constraint, and the omission of the **global reward** component. Our findings highlight that the full SRUM model achieves the highest overall accuracy, with the ablation results confirming the critical role of each component. The omission of the global reward led to a notable decrease in performance, underscoring its importance for capturing the overarching coherence and compositional structure of the generated images. While our findings highlight that the **KL constraint** is crucial for the model's performance, its removal resulted in a less severe drop, proving its value in ensuring training stability. Furthermore, using a simple **binarized reward** led to a significant performance degradation, which reinforces the necessity of a continuous and fine-grained reward signal to provide richer gradient information.

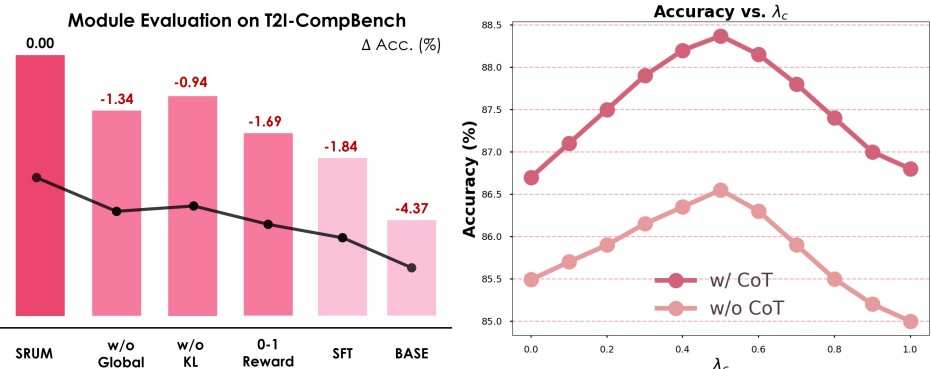

Figure 3: **Left:** Module Evaluation. We report the accuracy drop ($\Delta$ Acc. %) from our SRUM. Specifically, 0-1 Reward represents the sparse reward. **Right:** Hyperparameters Evaluation on T2I-CompBench. We report the accuracy in different $\lambda$ under two models: CoT and without CoT.

In the Figure 3 Right, we analyze the effect of different constraint ratios on the experimental outcomes. Across both Bagel with CoT and without CoT configurations, the results consistently indicate that $\lambda_c = 0.5$ is the most effective choice. Consequently, we set this hyperparameter as fixed one in our subsequent experiments for more significant evaluation results.

Finally, relying on a basic **sample-level reward** yielded the most significant performance drop among all variants, thereby validating that the complexity of the T2I-CompBench task demands a more holistic and comprehensive reward scheme. In conclusion, this systematic ablation study confirms that the efficacy of our proposed framework stems from the synergistic contributions of each reward component. This aligns with conclusions from post-training methods like Direct Preference Optimization (DPO) (Rafailov et al., 2023) where such a constraint is essential to prevent the model from significant policy deviation due to reward hacking. Additionally, we also explpore the binarized rewards like Dance-GRPO (Xue et al., 2025). We observed that this type of reward can underperform SFT and is ill-suited for regional feedback, which highlights the value of a dense reward structure.

**Further Analysis**. For a more granular investigation, we leverage the same powerful MLLM like QwenVL-2.5-72B from our primary evaluation to conduct a deeper analysis of our method and the baseline. Specifically, we employ the MLLM to perform a step-by-step scoring of the inference process. The evaluation is divided into two metrics: (1) layout, which assesses the concordance of the overall structure and quality, and (2) detail, which measures the fidelity of the generated fine-grained details. Our ablation study, visualized in Figure 4, systematically isolates the effects of each component. We observe that the "think" mode primarily bolsters the initial layout generation by improving the high-level reasoning process. The global reward component of SRUM then further refines this layout during the early stages of inference. In contrast, a baseline using only this global reward (labeled 'sample reward') yields negligible improvements in detail fidelity. This highlights a crucial finding: the fine-grained, local rewards are essential for the subsequent optimization of

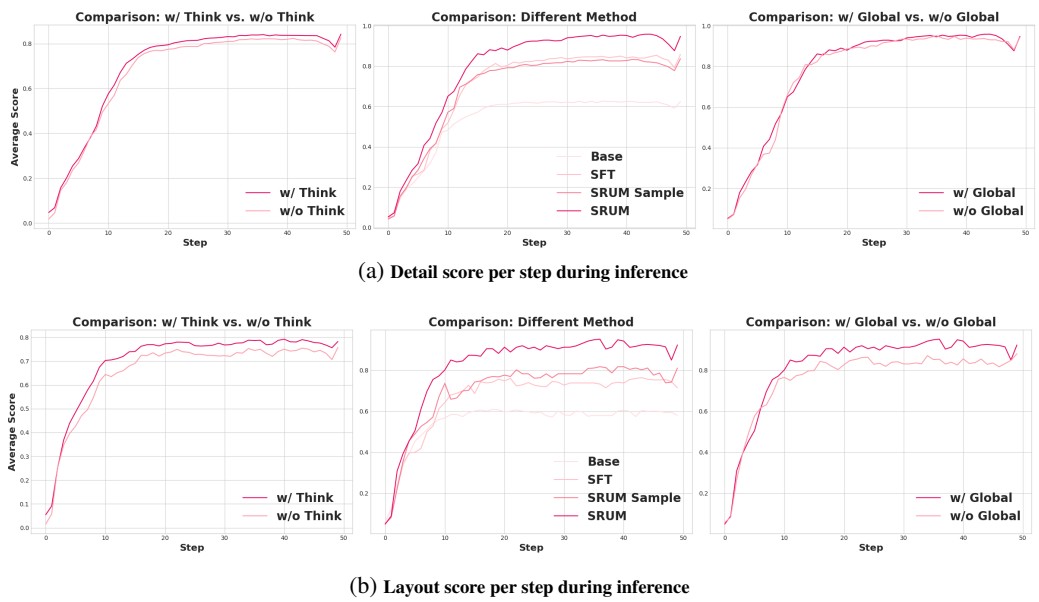

(a) **Detail score per step during inference**

(b) **Layout score per step during inference**

Figure 4: Score per step during inference in Bagel with its ablation models.

details, with their benefits becoming most apparent in the later inference steps. Collectively, these results demonstrate that our dual global-local reward mechanism provides a multi-stage optimization path: first establishing a coherent layout and then progressively refining the details. This synergistic approach allows SRUM to significantly outperform standard SFT on same self-generated data.

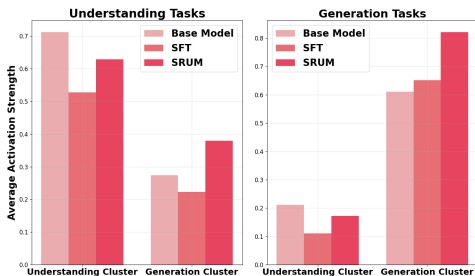

Figure 5: Functional cluster activation patterns of the different models (Bagel, SFT and SRUM) on understanding and generation tasks. The average activation strength of Understanding and Generation clusters is shown.

| | Base | SFT | SRUM |
|---|---|---|---|
| **MME-P** | 1687 | 1682 | 1673 |
| **MME-C** | 701 | 683 | 677 |
| **MMBench** | 85.0 | 84.6 | 84.8 |
| **MM-Vet** | 67.2 | 66.5 | 67.0 |
| **MMMU** | 55.3 | 55.0 | 55.2 |
| **MathVista** | 73.1 | 72.8 | 73.0 |
| **MMVP** | 69.3 | 68.7 | 70.0 |

Table 2: Comparison with the results of different models (Bagel, SFT and SRUM) on understanding benchmarks. MME-P and MME-C represents the perception and the cognition part respectively.

**Impact on Understanding Module**. Table 2 As shown in Table 2, our method has a minimal impact on the model's core understanding capabilities. On prevalent benchmarks such as MME (Fu et al., 2023), MM-Vet (Yu et al., 2024b), MMBench (Liu et al., 2024b), MMMU (Yue et al., 2024), and MathVista (Lu et al., 2023), the results exhibit only marginal fluctuations compared to the base version. Notably, performance on MMVP (Tong et al., 2024b) even improves which consistent with prior works (Tong et al., 2024a; Wang et al., 2024c;a). This indicates that our method holds significant potential for further iterative enhancement. In Figure 5, we track the activation dynamics of two distinct functional clusters, Understanding and Generation, across Base model, SFT and SRUM. In Bagel's inference, the mainstream parameters activated in the general understanding reasoning process are defined as the understanding cluster of parameters, and the mainstream parameters activated in the general generative reasoning process are defined as the understanding cluster of parameters. Our analysis reveals two distinct finetuning paradigms. Conventional SFT exhibits a narrowing effect, achieving specialization by suppressing irrelevant functional clusters. In contrast, our SRUM algorithm demonstrates an enhancing and orchestrating effect, strengthening the primary

task-relevant cluster while maintaining supportive activation in secondary clusters. This promotes robust and generalizable representations. Details setting can be seen in Appendix Section C.

**In-Domain Generalization**. We then investigate the in-domain generalization capability of our model. We posit that the compositional abilities learned from the T2I-CompBench training set should be transferable to other benchmarks with similar evaluation perspectives. To test this hypothesis, we evaluate our model—trained solely on T2I-CompBench—on the GenEval benchmark without any further fine-tuning. The comparative results are summarized in Table 3. As shown in the table, SRUM achieves strong performance across multiple attribute categories on GenEval. It obtains the highest scores in two key aspects: **Counting** (0.83) and **Color attr.** (0.83), outperforming both the base Bagel model and SFT. This indicates a robust understanding of numerical and color-based constraints, which are core to compositional reasoning. Although SFT excels in the **Colors** category (0.92), our method maintains competitive performance (0.90) while providing more balanced results across attributes. These results clearly demonstrate that the improvement enabled by our approach on this comparable benchmark is unequivocal. The model not only retains proficiency in simpler tasks such as single-object generation but also shows enhanced performance in more complex scenarios like object counting and color–attribute binding. This confirms strong in-domain generalization, affirming that improvements introduced in our method can transfer effectively to unseen data from a similar domain.

Table 3: Results on key visual attributes at GenEval. For brevity, some model names have been shortened. **Bold values** are the best in each column.

| Model | Single obj. | Two obj. | Counting | Colors | Position | Color attr. |
|---|---|---|---|---|---|---|
| Bagel | **0.99** | 0.94 | 0.81 | 0.88 | 0.64 | 0.82 |
| Bagel$_{+\text{SFT}}$ | 0.96 | 0.94 | 0.79 | **0.92** | 0.59 | 0.78 |
| Bagel$_{+\text{SRUM}}$ | 0.98 | 0.94 | **0.83** | 0.90 | 0.64 | **0.83** |

**Knowledge-based Generalization**. Following this, we explore whether our method holds a distinct advantage for the task of reasoning generation, a current area of focus in the community. Consequently, we designed an experiment wherein we train the model on one category of prompts from the WISE Benchmark and perform in-domain evaluations on the remaining two categories. This method allows us to construct three distinct evaluation sets for a thorough analysis of the model's generalization capabilities.

Table 4: The performance gain of the Bagel on unseen tasks after being trained on a specific domain. This table shows the percentage improvement in scores for the base model and the CoT model under different training/testing combinations.

| Training Domain | Spatio-temporal | | Natural science | | Common sense | |
|---|---|---|---|---|---|---|
| | Base | CoT | Base | CoT | Base | CoT |
| Common sense | +0.7% | **+6.0%** | **+2.3%** | +1.0% | — | — |
| Natural science | **+2.7%** | +4.0% | — | — | **+4.0%** | **+2.0%** |
| Spatio-temporal | — | — | +1.3% | **+2.0%** | +2.0% | +1.0% |

As illustrated in Table 4, selecting any single group for training universally enhances the image generation performance of the other two groups. This improvement is consistent across both standard and chain-of-thought (CoT) reasoning paradigms.

**Out-of-Domain Knowledge-based Generalization**. To further assess our model's generalization to unseen domains, we utilize T2I-ReasonBench, a large-scale and well-regarded benchmark for analyzing the reasoning quality of generated images. In this experiment, we take the model trained with the T2I-CompBench prompts and directly evaluate its performance on the benchmark. This setup is designed to demonstrate our model's out-of-domain generalization for advanced, reasoning-based image generation. Our primary focus is on accuracy-related scores, which measure the high-level semantic alignment between the model's output and the given prompt.

As illustrated in the Table 5, our SRUM method achieves a superior understanding of the given instructions compared to both the SFT and Base models. While SFT also yields a noticeable improvement, the enhanced performance of SRUM demonstrates that our approach effectively improves

Table 5: Detailed evaluation results of the Bagel model for different categories, combining accuracy (Acc.) and quality (Qual.) scores. **Bold values** represent the best performance in each column.

| Model | Entity | | Idiom | | Scientific | | Textual Image | |
|---|---|---|---|---|---|---|---|---|
| | Acc. | Qual. | Acc. | Qual. | Acc. | Qual. | Acc. | Qual. |
| Bagel | 36.9 | 88.1 | 29.7 | 77.3 | 40.2 | **69.5** | 40.49 | 71.5 |
| Bagel$_{+SFT}$ | 38.4 | 86.9 | 35.1 | 78.4 | 40.3 | 68.9 | 41.2 | 70.0 |
| Bagel$_{+SRUM}$ | **40.9** | **88.7** | **36.1** | **80.2** | **40.7** | 69.2 | **42.86** | **72.6** |

generalization on complex problems from both a data and an algorithmic perspective. Furthermore, in the evaluation of image-based instructions, SRUM provides consistent performance gains, in stark contrast to the volatility exhibited by the SFT model. This further substantiates that our algorithmic design is more adaptable, taking into account more nuanced factors than a SFT approach.

## 5 CONCLUSION

This paper introduces SRUM, a fine-grained post-training framework that enables a model's understanding module to reward its generation module. Addtionally , SRUM decomposes the reward into local and global components, facilitating multi-scale alignment and refinement. Extensive experiments validate SRUM's effectiveness, setting new state-of-the-art results on complex compositional and reasoning benchmarks such as T2I-CompBench and T2I-ReasonBench. The framework demonstrates robust in-domain and out-of-domain generalization, and our empirical analysis confirms the efficacy of the fine-grained reward design. These findings illuminate the synergistic development of understanding and generation capabilities within a single model and establish the principle of self-reward as a promising direction for future research.

## ETHICS STATEMENT

In accordance with the ICLR Code of Ethics, this work aims to contribute to society and human well-being by advancing fundamental knowledge in machine learning. Our research is primarily theoretical and was validated on publicly available, anonymized benchmark datasets. We are committed to upholding high standards of scientific excellence by presenting our methods and results in a transparent and reproducible manner, as detailed in our Reproducibility Statement.

We have considered the ethical principles outlined in the Code and do not foresee any direct negative consequences, risks to privacy, or potential for discrimination arising from this research. Our goal is the responsible stewardship of scientific inquiry, and we have been honest and transparent about the scope and limitations of our work.

## REPRODUCIBILITY STATEMENT

The source code for the SRUM framework will be made publicly available upon publication. To ensure the transparency and verifiability of our results, we have included detailed training logs for our main experiments in the supplementary materials. These logs provide a record of the training process and convergence, supporting the outcomes reported in our paper.

Furthermore, our work is based on publicly available models (Bagel and Blip30), and we have provided a comprehensive description of our methodology in Section Section 3.3. A complete training recipe, including all necessary hyperparameters for replication, is detailed in Appendix Section B. We believe that these materials offer a clear and sufficient basis for the community to reproduce our findings.

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

## A   USE OF LLM

We used a large language model (LLM) in a very limited capacity, restricted to minor editing of grammar, phrasing, and readability. The LLM was not involved in designing the method, developing theoretical results, or conducting experiments. All technical contributions, equations, and results are solely the work of the authors.

## B   DETAIL SETTINGS

Following the configuration of stage 4 from the **Bagel** (Deng et al., 2025) framework during our post-training phase, we employed the **AdamW** optimizer (Loshchilov, 2017), configured with momentum parameters $\beta_1 = 0.9$ and $\beta_2 = 0.95$. Drawing inspiration from (Molybog et al., 2023), we set the epsilon value to $1.0 \times 10^{-15}$ to mitigate loss spikes. When we increase the resolution during generation, we also adjust the diffusion timestep from 1.0 to 4.0, which helps maintain a stable noise-level distribution. We chose a constant learning rate, as this approach, as suggested by (Hu et al., 2024), simplifies the scaling of training data without needing to restart the training process. These empirical observations, along with established practices for large-scale model training (Goyal, 2017; Hoffmann et al., 2022; Kaplan et al., 2020), informed our final training protocol.

Our model architecture builds upon the standard Transformer (Vaswani et al., 2017) and Vision Transformer (ViT) (Dosovitskiy et al., 2021) paradigms, incorporating modern enhancements for stability and efficiency, such as Root Mean Square Layer Normalization (Zhang & Sennrich, 2019), GLU variants for activation functions (Shazeer, 2020), Rotary Position Embedding (RoPE) (Su et al., 2024), and Grouped-Query Attention (Ainslie et al., 2023). The generative process is fundamentally based on principles from Denoising Diffusion Probabilistic Models (DDPM) (Ho et al., 2020; Sohl-Dickstein et al., 2015), score-based modeling (Song et al., 2021), and utilizes classifier-free guidance (Ho & Salimans, 2022) within a latent space (Rombach et al., 2022) for high-resolution synthesis. The complete training recipe is summarized in Table 6.

Table 6: **Training recipe of SRUM.**

| Hyperparameters | Post-training |
|---|---|
| Learning rate | $2.5 \times 10^{-5}$ |
| LR scheduler | Constant |
| Weight decay | 0.0 |
| Gradient norm clip | 1.0 |
| Optimizer | AdamW ($\beta_1 = 0.9$, $\beta_2 = 0.95$, $\epsilon = 1.0 \times 10^{-15}$) |
| Warm-up steps | 500 |
| Max context window | 40k |
| Gen resolution (min short side, max long side) | (512, 1024) |
| Diffusion timestep shift | 4.0 |

In Section 3.1, we explain how to generate detection boxes in all cases. Here, we note that Bagel uses an external model (SAM), while BLIP3o relies on its own native capabilities. We suggest that the rationale for this choice can be based on the model's performance on grounding benchmarks (such as RefCOCO).

## C   DEFINITION AND CALCULATION OF AVERAGE ACTIVATION STRENGTH

To investigate the internal functional mechanisms of different training methods, we introduce the metric of *Average Activation Strength*. This metric is designed to quantify the overall activity level of a predefined functional neural cluster when the model is performing a specific type of task. This appendix provides a detailed definition, mathematical formulation, and the statistical implementation procedure. The **Average Activation Strength** is defined as the mean activation value of all neurons within a specific functional cluster, averaged over an entire dataset for a given task. The calculation involves a two-level averaging process:

1. **Intra-Cluster Average:** For a single input sample, we compute the mean of the activation values of all neurons belonging to the target cluster.

2. **Dataset-Wide Average:** We then average these single-sample cluster means across all samples in the entire task dataset.

This metric reflects the degree of engagement of a functional cluster (e.g., the "Understanding Cluster") while processing a certain category of tasks (e.g., "Generation Tasks"). A higher value indicates that the cluster is more strongly and broadly activated for that task.

To formalize this definition, we first introduce the following notation:

- $M$: A specific neural network model (e.g., Base, SFT, or SRUM).
- $C_k$: A functional neural cluster $k$ (e.g., $C_{understand}$ or $C_{generate}$), which is a set of specific neuron indices.
- $|C_k|$: The number of neurons in cluster $C_k$.
- $D_T$: The dataset for a specific task type $T$ (e.g., $D_{understanding}$ or $D_{generation}$).
- $|D_T|$: The number of samples in the dataset $D_T$.
- $x$: An individual input sample from the dataset, where $x \in D_T$.
- $a_i(x)$: The activation value of neuron $i$ in model $M$ given the input $x$, where $i \in C_k$. This typically refers to the output of a neuron after its activation function (e.g., ReLU or GeLU) has been applied.

For a single input sample $x$, the average activation strength of a cluster $C_k$, denoted as $S_{sample}$, is calculated as:

$$S_{sample}(M, C_k, x) = \frac{1}{|C_k|} \sum_{i \in C_k} a_i(x) \tag{4}$$

The final **Average Activation Strength** of cluster $C_k$ for model $M$ over the entire dataset $D_T$, denoted as $S_{final}$, is the expected value of $S_{sample}$ over all samples. In practice, this is estimated by averaging across the dataset:

$$S_{final}(M, C_k, D_T) = \frac{1}{|D_T|} \sum_{x \in D_T} S_{sample}(M, C_k, x) = \frac{1}{|D_T||C_k|} \sum_{x \in D_T} \sum_{i \in C_k} a_i(x) \tag{5}$$

This $S_{final}$ value corresponds to the height of each bar in the activation figures. Algorithm details can be seen in Algorithm 1.

## D   DATA CURATION

We leverage the training instructions from T2I-CompBench (Huang et al., 2023) to guide our image generation process. Specifically, we utilize the generation capabilities of UMs (Wu et al., 2024b;a; Xie et al., 2024; Dong et al., 2024), which are representative of the state-of-the-art in text-to-image synthesis (Betker et al., 2023; Saharia et al., 2022; Esser et al., 2024; Labs, 2024; Wu et al., 2025), to synthesize corresponding images based on these instructions. Subsequently, the understanding end of UMs, which possesses powerful vision-language comprehension abilities akin to models like LLaVA, InternVL, and Gemini (Liu et al., 2024a; Chen et al., 2024b; Wang et al., 2024b; Team et al., 2023), is employed to evaluate and score the generated images.

The capabilities of these models are built upon massive web-scale datasets (Schuhmann et al., 2022; Li et al., 2024a) and canonical vision datasets (Lin et al., 2014), which are often enhanced with high-quality captioning and instruction-following data (Sharma et al., 2018; Li et al., 2024b; Liu et al., 2023a). Our prompting strategy for eliciting rewards is inspired by the methodologies used in instruction-based image editing (Brooks et al., 2023; Wei et al., 2024; Zhang et al., 2023a; Yu et al., 2024a; Hui et al., 2024; Bai et al., 2024). The detailed data used in this evaluation are as follows:

**Algorithm 1** Calculation of Average Activation Strength (with Single Threshold-Based Cluster Definition)

1: **Step 1: Define Neuron Clusters**
2:   **Require:**
3:     $M_{\text{base}}$: Base model.
4:     $D_{\text{und}}$: Understanding dataset.
5:     $D_{\text{gen}}$: Generation dataset.
6:     $\tau_{\text{act}}$:  Activation percentile threshold (%).
7:   **Ensure:** $C_{\text{understand}}, C_{\text{generate}}$.
8:
9:     *// (1.1) Collect mean activations*
10:    Let $N$ be set of FFN neurons.
11:    Init maps $\mu_{\text{und}}, \mu_{\text{gen}}, \mu_{\text{max}}$.
12: **for** each neuron $n \in N$ **do**
13:      $\mu_{\text{und}}[n] \leftarrow \text{mean}_{x \in D_{\text{und}}} a_n(x)$
14:      $\mu_{\text{gen}}[n] \leftarrow \text{mean}_{x \in D_{\text{gen}}} a_n(x)$
15: **end for**
16:
17:    *// (1.2) Calculate max activation and threshold*
18: **for** each neuron $n \in N$ **do**
19:      $\mu_{\text{max}}[n] \leftarrow \max(\mu_{\text{und}}[n], \mu_{\text{gen}}[n])$
20: **end for**
21: $V_{\text{act}} \leftarrow \text{Percentile}(\{\mu_{\text{max}}[n] \mid n \in N\}, \tau_{\text{act}})$
22:
23:    *// (1.3) Filter clusters based on activation threshold and max activation task*
24:    $C_{\text{understand}} \leftarrow \emptyset, C_{\text{generate}} \leftarrow \emptyset$
25: **for** each neuron $n \in N$ **do**
26:    **if** $\mu_{\text{max}}[n] \geq V_{\text{act}}$ **then**       ▷ Must be an active neuron
27:      **if** $\mu_{\text{und}}[n] > \mu_{\text{gen}}[n]$ **then**       ▷ More active for understanding
28:        $C_{\text{und}} \leftarrow C_{\text{und}} \cup \{n\}$
29:      **else if** $\mu_{\text{gen}}[n] > \mu_{\text{und}}[n]$ **then**       ▷ More active for generation
30:        $C_{\text{gen}} \leftarrow C_{\text{gen}} \cup \{n\}$
31:      **end if**
32:    **end if**
33: **end for**
34:    *// Clusters fixed*

1: **Step 2: Prepare Eval Data & Model**
2:    Prepare dataset (e.g., $D_{\text{und}}$).
3:    Load model $M$.
4:
5: **Step 3: Forward Pass & Log**
6:    Init lists: und_activ = [], gen_activ = []
7: **for** each sample $x \in D_{\text{understanding}}$ **do**
8:    Forward pass $M(x)$.
9:    Record $a_i(x)$ for $i \in C_{\text{und}}, C_{\text{gen}}$.
10:   Calc sample avg activation:
11:      $S_{\text{samp, und}} \leftarrow \text{mean}_{i \in C_{\text{und}}} a_i(x)$
12:      $S_{\text{samp, gen}} \leftarrow \text{mean}_{i \in C_{\text{gen}}} a_i(x)$
13:   Append $S_{\text{samp, und}}$ to und_activ
14:   Append $S_{\text{samp, gen}}$ to gen_activ
15: **end for**
16:
17: **Step 4: Final Aggregation**
18:   $S_{\text{final, und}} \leftarrow \text{mean}(\text{und\_activ})$
19:   $S_{\text{final, gen}} \leftarrow \text{mean}(\text{gen\_activ})$
20:   Output $S_{\text{final, und}}, S_{\text{final, gen}}$.
21:
22: **Step 5: Repeat Process**
23:   Repeat Steps 3-4 using $D_{\text{gen}}$.
24:   Repeat Steps 2-5 for each model.

**Generated Prompt Content:**

```
# TASK: Global Layout and Composition Analysis
You are an expert image analyst.

Your task is to score the overall composition
of an image based on a user's prompt. Focus solely
on how the arrangement of elements and scene structure
align with the prompt's spatial intent.

**Original Prompt:** "{original_prompt}"
---
## YOUR TASK & OUTPUT FORMAT
Provide a single score from **-1.0 to 1.0** and a brief reason.

* **Scoring Guide:**
* **1.0:** Perfect alignment with the prompt's
spatial intent.
* **0.5 to 0.9:** Mostly correct layout
with minor flaws.
* **-0.4 to 0.4:** Neutral. No specific spatial
info in prompt, or generic layout.
* **-0.9 to -0.5:** Incorrect layout or
contradictory to the prompt.
* **-1.0:** Fundamentally contradicts the
prompt's spatial intent.

* **Output Lines:**
    'Score: [A single number between -1.0 and 1.0]'
    'Reason: [Your justification]'
---
## DIVERSE EXAMPLES

### Example 1 (Perfect Alignment)
Score: 0.95
Reason: The wide shot of a sunset over the ocean perfectly
matches the prompt's implied composition.

### Example 2 (Contradictory Layout)
Score: -0.7
Reason: The cat is on the right of the dog, but the prompt
asked for the cat on the left.
---
Begin your analysis now.
```

Table 7: **Documentation for `create_global_layout_reward_prompt`.**

**Generated Prompt Content:**

```
# TASK: Integrated Region Analysis and Scoring
You are an expert AI image analyst.
Your task is to analyze unlabeled regions in an image
based on a user's prompt.
For each region, you will perform a two-stage analysis.

**Original Prompt:** "{original_prompt}"
---
**UNLABELED REGIONS FOR YOUR ANALYSIS:**
{regions_text}
---
## YOUR TWO-STAGE TASK & OUTPUT FORMAT
For **every Region ID** listed above,
you must perform the following steps.

### STAGE 1: Identify Object
First, identify the primary object within the bounding box.
* **Output Line:**
'Identified Object: [Your description of the object]'

### STAGE 2: Score and Justify
Provide a single, overall score
from **-1.0 to 1.0** that considers BOTH the object's
**relevance** to the prompt and its **visual quality**.
You must provide a clear reason for your score.
Be as strict as possible and only give full marks
when the image quality is beyond doubt.

* **Scoring Guide:**
    * **1.0:** Perfect. The object is exactly what the
    prompt asks for and is technically flawless and perfect.
    * **0.5 to 0.9:** Very good. A highly relevant object
    with minor flaws, or a well-executed secondary element.
    * **-0.4 to 0.4:** Neutral/Acceptable. A moderately
    relevant object, an object with mixed qualities, or an
    irrelevant but harmless background element.
    A score of 0.0 is perfectly neutral.
    * **-0.9 to -0.5:** Bad. The object is irrelevant
    and distracting, or it is a relevant object with
    severe visual artifacts/flaws.
    * **-1.0:** Very Bad. The object actively
    undermines the image and directly
    contradicts the prompt's intent.

* **Output Lines:**
    'Score: [A single number between -1.0 and 1.0]'
---
## EXAMPLE OUTPUT STRUCTURE
**Region ID: 1**
Identified Object: A running golden retriever.
Score: 0.95
---
**Region ID: 2**
Identified Object: A tall green tree in the background.
Score: 0.2
---
Begin your analysis now.
```

Table 8: **Documentation for `create_hybrid_evaluation_prompt`.**

Table 9: VLM Rewards for Prompt: "a microwave hidden by a horse"

| Object | Bounding Box (bbox) | Score | Reason |
|---|---|---|---|
| global_layout_reward | [0, 0, 1024, 1024] | 1.00 | The image perfectly aligns with the prompt's spatial intent by depicting a horse positioned in front of a microwave, effectively hiding it from view. The composition is well-executed, with the horse's body and legs obscuring the microwave, and the plain background ensuring focus on the interaction between the two elements. |
| A brown horse with a white blaze and white socks. | [164, 97, 957, 990] | 0.95 | – |
| A brown horse with a white blaze and white socks. | [0, 0, 1023, 831] | 0.95 | – |
| A brown horse with a white blaze and white socks. | [349, 28, 920, 880] | 0.95 | – |
| A microwave. | [349, 28, 920, 389] | 0.50 | – |
| The floor. | [0, 681, 1023, 1023] | 0.00 | – |
| The floor. | [0, 838, 1023, 1023] | 0.00 | – |
| A brown horse with a white blaze and white socks. | [422, 94, 748, 292] | 0.95 | – |
| A brown horse with a white blaze and white socks. | [429, 589, 856, 795] | 0.95 | – |
| A brown horse with a white blaze and white socks. | [430, 121, 848, 793] | 0.95 | – |
| A brown horse with a white blaze and white socks. | [430, 607, 755, 780] | 0.95 | – |

## E   FAILURE CASES STUDY

We conducted an analysis of three failure cases:

1. **The language model is unable to arrive at the correct answer.** Our prompt was:

   "Given the following mapping: 1 – apple, 2 – banana, 3 – watermelon. Compute: $1 + 3 - 2 + 1$, then return the fruit corresponding to the result."

   In this scenario, most language models answer incorrectly. Therefore, the generation module in this case can only generate "apple."

2. **Causal multi-image generation.** Because the training data for Bagel rarely contains data representing causality in a single image, we are unable to achieve good results for this type of task. Our example was:

   "Generate a comparison image of British cities before and after the Industrial Revolution."

3. **Aesthetic generation issues.** Our method focuses on problems related to reasoning, knowledge, and composition. Consequently, aesthetics are not a primary consideration, which is also a common issue in existing models. Our example was:

   "Generate a particularly beautiful chair."

The top row shows our failure cases, and the bottom row shows the failure cases of nano-banana (current frontier model), illustrating that this failure is a systemic problem in generative models.

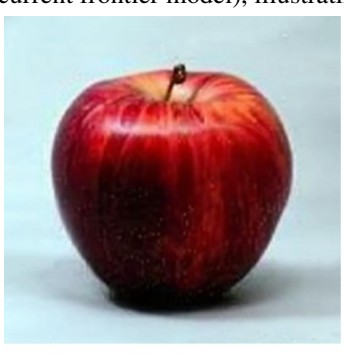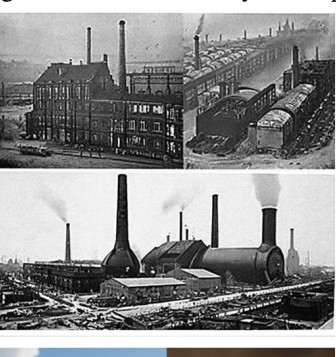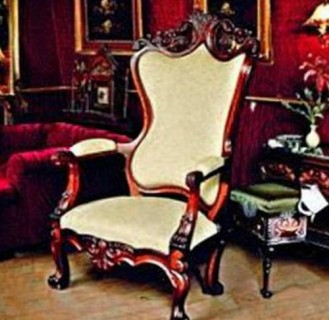
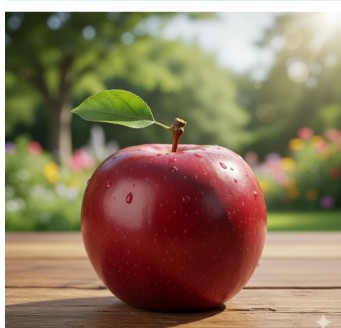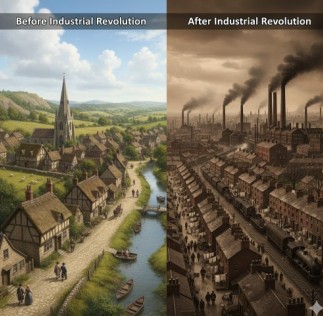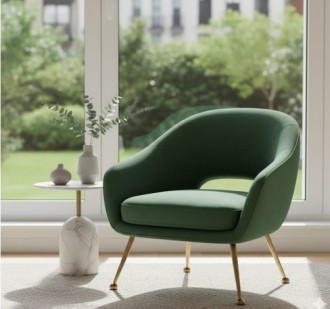

