# OpenReview forum: "SRUM: Fine-Grained Self-Rewarding for Unified Multimodal Models"
_ICLR.cc/2026/Conference — ICLR 2026 Conference Withdrawn Submission_

### Official Review · Reviewer_PxGQ · 2025-10-28

**Soundness:** 3
**Presentation:** 3
**Contribution:** 3
**Rating:** 4
**Confidence:** 4

**Summary:**

This paper proposes SRUM, a self-rewarding method for unified multimodal models. It lets the model’s own understanding part judge and guide its image generation, so it can better match prompts without using human feedback. The method gives both global and local rewards to improve structure and details in generated images. Experiments show clear improvement on several benchmarks, meaning SRUM helps the model connect understanding and generation more effectively.

**Strengths:**

1. The design of global and local rewards gives detailed and effective guidance, which can improve both scene structure and object details in generation.
2. The experiments are thorough and comprehensive in terms of the coverage of different backbones, the ablations, and the generalization analysis with diverse benchmarks.

**Weaknesses:**

1. The paper mainly compares SRUM with the raw base models instead of other post-training or reward-based approaches. This makes the improvement less convincing and does not clearly show advantages over existing post-train methods.
2. Although claimed as self-rewarding, the evaluation still relies on strong external MLLMs (e.g., QwenVL2.5-72B) for scoring, so the method is not truly self-evaluative as described.
3. The pipeline involves multiple stages (reward generation, bounding box detection, and reward-weighted training) with many hyperparameters, which may limit reproducibility and accessibility.

**Questions:**

1. Why not compare with other methods that use RL to boost image generation?
2. Can you provide visual case studies or detailed analysis of failure modes?

---

> ### Author Response · Authors · 2025-11-17
>
> **To Reviewer PxGQ:**
>
> Thank you for your reply. We acknowledge that you have observed some areas that we did not claim with absolute clarity. Therefore, we hope to address your concerns with the following responses:
>
> **W1: Comparison with Other Post-Training Methods** We have conducted comparative experiments between our method and the popular post-training method, ReCA. The results demonstrate that our method holds a significant advantage (please see the table for detailed data).
>
> | Model       | Configuration | Score |
> | :---------- | :------------ | ----: |
> | UnifiedReward   | -             | 85.15 |
> | ReCA  | -             | 86.45 |
> | **SRUM**  | -             |88.37 |
>
> Current research \[1\] indicates that UnifiedReward is considered one of the optimal reward modeling schemes within mainstream RL post-training paradigms like DPO and GRPO. However, under our experimental setup, its performance was far inferior to our proposed SRUM framework. This comparison fully illustrates that our method has a clear and solid advantage in effectiveness over traditional RL post-training paths.
>
> **W2: Clarification on the Use of External Models** We wish to clarify here: the external 72B model cited in the paper **was used only during the evaluation phase**. Its function is equivalent to an evaluation metric to measure our algorithm's performance, similar in role to Accuracy (Acc) in an ImageNet classification task or mean Intersection-over-Union (mIOU) in a segmentation task.
>
> In our training and core algorithmic pipeline, we **do not rely on any external models whatsoever**. All reward signals are generated by our own model's understanding module and are used for self-optimization.
>
> **W3: Hyperparameter Analysis** In our method's design, the most critical hyperparameter is the specific value of the **reward weight** during the training process. We have added a systematic analysis of this weight in the revised manuscript (please see the right-hand subplot in Figure 3). As for the other components in the pipeline, such as the reward generation and bounding box generation steps, we maintained their default configurations to ensure the method's simplicity and reproducibility, facilitating direct use and validation for future research.
>
> **Q1: Comparison with RL Post-training** Please see the explanation in W1.
>
> **Q2: Evidence of Model Limitations** See **Appendix Section E**, where we provide relevant case studies and explain that these problems are not caused by our algorithm but are some fundamental issues of the generative model.
>
> If you are satisfied with our response, please consider updating your score. If you need any clarification, please feel free to contact us.
>
> \[1\] Tong, C., Guo, Z., Zhang, R., Shan, W., Wei, X., Xing, Z., ... & Heng, P. A. (2025). Delving into RL for Image Generation with CoT: A Study on DPO vs. GRPO. arXiv preprint arXiv:2505.17017.

---

> > ### Comment · Reviewer_PxGQ · 2025-11-27
> >
> > Thanks for the clarification from the authors, I will increase my score.

---

> > > ### Author Response · Authors · 2025-11-27
> > >
> > > Thanks for the score improvement. Your opinions have greatly improved the quality of our manuscripts!

---

### Official Review · Reviewer_bnfL · 2025-10-29

**Soundness:** 3
**Presentation:** 3
**Contribution:** 3
**Rating:** 4
**Confidence:** 4

**Summary:**

This paper introduces SRUM, a framework designed to improve the alignment between the understanding and generation modules of Unified Multimodal Models (UMMs) for text-to-image (T2I) tasks. ​ The process involves two key stages: training data creation and post-training refinement.

In the first stage, SRUM generates high-quality image candidates and bounding boxes using the UMM’s generation module, which are then evaluated by its understanding ability. ​ The understanding module acts as an internal evaluator, providing dual-level feedback through global rewards for compositional structure and local rewards for object-level fidelity. ​
In the second stage, these rewards are used in a reward-weighted post-training process, where the generation module is refined based on the feedback. ​ This self-rewarding loop enables the model to improve its generative capabilities without relying on external models or human-labeled data. ​

Experiments show that SRUM significantly enhances image accuracy, compositional reasoning, and generalization across benchmarks, setting new state-of-the-art results and establishing a novel paradigm for self-improvement in UMMs.

**Strengths:**

The problem is good and fundemental: the gap between understanding and generation of a unified model. and post training seems a promising direction. also their approach is self-contained  which eliminates the need for external models or human-labeled data, making the approach more efficient and scalable.

Dual-Level Reward System: SRUM employs a two-part reward system—global rewards for overall compositional structure and local rewards for fine-grained object-level fidelity. ​ This multi-scale feedback mechanism is critical for improving complex image generation tasks.


Good performance and generalization: The framework achieves significant improvements in image accuracy and compositional reasoning, setting new benchmarks on T2I-CompBench and T2I-ReasonBench. It also demonstrates strong generalization

**Weaknesses:**

Two main concerns:

1, why choose to use the model itself as understanding model? It is nice and elegant which remove need of external VLM, but is there a real reason to have to use itself? I feel this paper lacks depth to show why it has to be itself

2, the approach is too biased towards composition, which can be solved in bbox. what if the misalignment is more global? say style, or object 3d angle?

i am willing to raise my score if authors can address my first concern

**Questions:**

see weekness

---

> ### Author Response · Authors · 2025-11-17
>
> **To Reviewer bnfL:**
>
> Thank you for your feedback. Your comments have been highly beneficial in addressing the core issues of our manuscript. In response, we have conducted supplementary experiments and wish to further clarify the related points.
>
> **W1: On the Necessity of Self-Rewarding vs. External Models** As the results for the Qwen series models in our experiments show, the introduction of external reward models **did not bring significant improvements** to our method. Furthermore, their effectiveness remained **highly dependent on the algorithm's internal design** (e.g., the presence of the global reward).
>
> Looking further at the experiments scaling from 7B to 32B, the external models did not have a decisive impact on the results. This finding is quite insightful: it indirectly confirms that the effectiveness of the SRUM method is rooted in its intrinsic design rather than reliance on external resources.
>
> | Model       | Configuration | Score |
> | :---------- | :------------ | ----: |
> | QwenVL-7B   | without global| 85.78 |
> | QwenVL-7B   | with global   | 87.52 |
> | QwenVL-32B  | without global| 85.53 |
> | QwenVL-32B  | with global   | 88.01 |
> | **SRUM**  | -             |88.37 |
>
> Our understanding module itself is "plug-and-play" and possesses the potential for future expansion into online learning. From an engineering deployment perspective, our method is also simpler, requiring only a single model to be deployed rather than multiple external modules. Therefore, this series of experimental phenomena further supports the necessity and rationale of the "self-rewarding" mechanism we advocate for.
>
> **W2: Limitations of Bounding Boxes and the Role of Global Reward** Your concern is highly relevant. In our initial design phase, we observed the same problem: relying solely on bounding boxes indeed has blind spots that are difficult to cover.
>
> To address this, we introduced the **global reward as an effective supplementary mechanism**. It is better equipped to handle factors that fall outside the scope of pure composition, such as generated content involving factual knowledge.
>
> We completely agree that how to further refine the bounding box design and optimize the local reward will be a crucial research direction for the future. The current global reward mechanism is an important first step toward that goal. **Tables 4 and 5** both demonstrate that our design is effective, especially from the perspective of not only compositional but complex (knowledge and reasoning) generation.
>
> If you are satisfied with our response, please consider updating your score. If you need any clarification, please feel free to contact us.

---

### Official Review · Reviewer_8hSG · 2025-10-31

**Soundness:** 3
**Presentation:** 3
**Contribution:** 2
**Rating:** 4
**Confidence:** 4

**Summary:**

The paper introduces SRUM, a post-training framework for unified multimodal models. The authors design a self-improvement to leverage the model's own powerful understanding module as an internal evaluator to provide corrective feedback to its generation module.

**Strengths:**

1. This paper introduces a post-training method for unified multi-modal models that requires no new human-labeled data.

2. The results on T2I benchmarks are good, ablations studies demonstrates considerable performance gains on BAGEL and BLIP3o, proving the effectiveness of the method.

**Weaknesses:**

1. The paper needs more proof on whether self-rewarding is generalizable to other unified multi-modal models. The effectiveness of the verifier is the key to the validity of SRUM, but the persistence of verifier quality is validated only on architectures (Bagel, Blip3o) that has a substantial degree of separation between their understanding and generation components. For more deeply integrated architectures (e.g., Emu3[1], NextStep-1[2]) where understanding and generation are entangled in the same core parameters, it is probable that generative post-training would lead to catastrophic forgetting and degradation of the model's understanding capability, causing verifier collapse. The paper lacks evidence that SRUM can prevent this mutual corruption. Which leads to my next concern:

2. The paper needs more justification on using UMM's internal understanding as verifier. Since the supervision is single directional (understanding -> generation), and there seems to be no mutual enhancement effect. Then **why not simply use an external expert (a strong VLM) as verifier**? Empirically, dedicated VLMs are usually stronger in perception and visual understanding than UMMs, and would be immune to potential verifier collapse as stated in the previous question.

[1] Wang, et. al., Emu3: Next-Token Prediction is All You Need. In arXiv.org: Vol. abs/2409.18869.

[2] Han, et al., NextStep-1: Toward Autoregressive Image Generation with Continuous Tokens at Scale. In arXiv.org: Vol. abs/2508.10711.

**Questions:**

See weakness.

---

> ### Author Response · Authors · 2025-11-17
>
> **To Reviewer 8hSG:**
>
> Thank you for your review. We understand your concern regarding the potential for **model collapse** in self-rewarding models, which has long been a difficult challenge in model self-evolution. We hope to engage in a more meaningful discussion with you through the following explanations:
>
> **W1: Mode Collapse in Deep Fusion Architectures** We would like to offer a slight clarification here: BLIP-3o and Bagel differ in their architectural approaches. **Bagel is a representative work of deep fusion(***Original text: “it offers a significant advantage by maintaining a bottleneck-free context throughout all transformer blocks, thereby enabling lossless interaction between the generation and understanding modules and is more amenable to scaling.”***).** This distinction is clearly defined in Bagel's original paper and in subsequent model classifications by the community \[1,2\], which is also why we selected these two representative frameworks for our experiments.
>
> Currently, the **Bagel** architecture is widely recognized as the **mainstream** model due to its exceptional performance and unified framework. This is the **primary reason** we selected Bagel as our core architecture and chose to apply the **SRUM** approach to it. In contrast, we decided to temporarily forego earlier **AR** models, primarily due to **computational resource demands** and **architectural** issues when integrating SRUM. Specifically, **NextStep** has not released its full training code, and **EMU-3** uses separate weights for its understanding and generation versions. However, we still add them to the relative discussion in **updated related works** for reference.
>
> Meanwhile, regarding whether the understanding module suffers from mode collapse, we demonstrated in **Table 2 of the original paper** that our SRUM algorithm does not encounter this issue, even within a deep fusion architecture like Bagel.
>
> **W2: On External Reward Models and Design Rationale** As shown in the table, the introduction of external reward models (including large-scale VLMs, , our experiments all used the same initial prompt data as SFT/SRUM) **did not bring significant improvements** to our method. Furthermore, their effectiveness remained **highly dependent on our own algorithmic design** (e.g., the inclusion of the global reward). This finding is somewhat counter-intuitive and does not fully align with conventional experience.
>
> | Model       | Configuration | Score |
> | :---------- | :------------ | ----: |
> | QwenVL-7B   | without global| 85.78 |
> | QwenVL-7B   | with global   | 87.52 |
> | QwenVL-32B  | without global| 85.53 |
> | QwenVL-32B  | with global   | 88.01 |
> | **SRUM**  | -             |88.37 |
>
> As a **self-rewarding** method, our algorithm's design is indeed centered on generation. This design motivation stems from the observation presented in the **Figure 1 teaser: the understanding capabilities of current UMMs far exceed their generation capabilities.** Therefore, we prioritized leveraging the strengths of the understanding module to drive improvements in the generation module, aiming to first bring both to a relatively balanced, high level of performance before further exploring paths for mutual enhancement.
>
> Furthermore, utilizing the internal module for self-reward holds greater potential for future algorithm designs, such as online learning and deeper fusion architectures. The workflow framework proposed in this work aims to provide a viable paradigm reference for subsequent research, and we hope it will reignite the community's interest in and exploration of self-rewarding mechanisms in UMMs. Furthermore, our primary motivation stems from the significant disparity in current UMMs, where comprehension capabilities substantially outpace generation capabilities. Therefore, our immediate priority is to leverage this advanced comprehension to enhance the generation module, aiming to bridge this gap and elevate the model's generative performance to a more desirable standard.
>
> If you are satisfied with our response, please consider updating your score. If you need any clarification, please feel free to contact us.
>
> \[1\] Deng, C., Zhu, D., Li, K., Gou, C., Li, F., Wang, Z., ... & Fan, H. (2025). Emerging properties in unified multimodal pretraining. arXiv preprint arXiv:2505.14683.
>
> \[2\] Xie, J., Darrell, T., Zettlemoyer, L., & Wang, X. (2025). Reconstruction alignment improves unified multimodal models. arXiv preprint arXiv:2509.07295.

---

### Official Review · Reviewer_YiSP · 2025-11-01

**Soundness:** 3
**Presentation:** 2
**Contribution:** 2
**Rating:** 6
**Confidence:** 3

**Summary:**

This paper introduces SRUM (Self-Rewarding for Unified Multimodal Models), a post-training framework that leverages a model's understanding module to improve its generation capabilities. The authors observe that UMMs possess strong understanding capabilities that significantly outperform their generation quality—a model can identify when a generated image doesn't match a prompt but cannot generate the correct image. SRUM addresses this gap by using the understanding module as an internal evaluator to provide fine-grained feedback during training. The method decomposes rewards into two components: (1) a global reward assessing overall compositional structure and (2) local rewards providing object-level feedback on specific image regions identified through bounding boxes. Training uses a reward-weighted velocity prediction loss combined with a reference loss to prevent reward hacking. The method achieves significant improvements on T2I-CompBench (82.18→88.37) and T2I-ReasonBench (40.7→50.4 in accuracy) without requiring additional human annotations.

**Strengths:**

1. The paper clearly identifies a genuine gap in UMMs—the disconnect between understanding and generation capabilities—and proposes an elegant self-rewarding solution that exploits the model's own capabilities without external reward models or new human labels.
2. The decomposition into global (compositional) and local (object-level) rewards is well-motivated and technically sound. This multi-scale feedback addresses the limitations of holistic scoring for complex compositional tasks.
3. There are substantial improvements across multiple benchmarks using SRUM method and strong generalization to in-domain (GenEval, WISE) and out-of-domain (T2I-ReasonBench) tasks, while there are minimal impact on understanding capabilities (Table 2).

**Weaknesses:**

1. The paper provides no analysis of computational costs, which is critical for evaluating practicality. Since the SRUM needs to use UMM to generate the self-rewards used for the training, the additional computation cost should be analyzed and compared to the normal method SFT.
2. The reward system may relies heavily on carefully engineered prompts (Tables 7-8 in Appendix). It's unknown how reliable these generated rewards are in a quantative view. For example, maybe using another VLM for scoring can bring more improvements.
3. Some abaltion experiments are missing. For example, the experiments using only the global reward as the reward without local reward. And the training with some external rewards like ImageReward Models? If these external rewards are better, then the self-rewarding part of the paper may be not necessary.

**Questions:**

1. Can you clarify clearly about the overall training process? From my understanding, you are generating candidate images, using UMM to generate rewards on the whole training dataset and then use it for the training. This seems to be a offline method. Have you considered online method that using the latested updated UMM to generate scores as rewards?
2. Can you provide results for "w/o Local" (using only global reward without object-level rewards)? This is essential to validate the necessity of fine-grained local feedback.
3. How robust is the method to variations in the reward prompts (Tables 7-8)? Were these prompts tuned on a validation set?
4. How does SRUM compare to:
- Using external reward models?
- Simple reconstruction alignment (RecA)?

Typos:
1. line 258: "modes" -> "models"
2. line 272: "Table Table 1" -> "Table 1"

---

> ### Author Response · Authors · 2025-11-17
>
> To Reviewer YiSP:
>
> Thank you for your valuable suggestions regarding our comparison methods and the need for further detailed experiments. We also appreciate you pointing out the typos, which we will correct in the revised paper.
> We address your specific weaknesses and questions point-by-point below:
>
> **W1: Cost Calculation**
>
> Our scoring process is, in fact, extremely efficient. By parallelizing the understanding module, we can complete the scoring task for a large batch of data in a very short amount of time. For instance, inference on 6,000 data samples requires no more than 4 GPU-hours on H100 GPUs. Concurrently, the loss calculation introduces negligible additional time or computational overhead (Training time: SFT: 12.4 GPU hours; SRUM: 12.5 GPU hours). More importantly, our self-rewarding mechanism eliminates the previously laborious manual data cleaning process. This is particularly crucial in the current context where high-quality image-text data is increasingly scarce.
>
> **W2: Prompt Design and External VLMs**
>
> The primary role of prompt design is to standardize the output format of the scores; it is not strongly correlated with the scoring quality itself. Without a reasonable prompt design to unify the output format, repeated filtering and adjustments would be necessary to obtain the required structured results.
>
> Furthermore, we conducted comparative experiments with external VLMs (e.g., Qwen-7B and Qwen-32B, as shown in the table, our experiments all used the same initial prompt data as SFT/SRUM.). The results indicate that introducing external models did not yield stable improvements compared to our method, and even showed a fluctuating decline, especially with the larger model. Critically, such minor gains are easily overshadowed by other algorithmic improvements. For example, after removing the global reward mechanism, the performance drop on the QwenVL model was far more significant than the change observed in our model's understanding module .
>
> | Model       | Configuration | Score |
> | :---------- | :------------ | ----: |
> | QwenVL-7B   | without global| 85.78 |
> | QwenVL-7B   | with global   | 87.52 |
> | QwenVL-32B  | without global| 85.53 |
> | QwenVL-32B  | with global   | 88.01 |
> | UnifiedReward   | -             | 85.15 |
> | ReCA  | -             | 86.45 |
> | **SRUM**  | -             |88.37 |
>
> **W3: External Reward Models and Local Reward Ablation**
>
> Regarding ablation studies and comparison with external reward models, we first tried introducing the popular external reward model UnifiedReward, but its performance was not ideal (please see the table in W2).
> Simultaneously, we conducted an ablation study by removing the local reward. The results showed that in CoT (Chain-of-Thought) mode, performance dropped by 0.76; in the standard inference mode, the drop was 1.04. For a detailed step-level analysis, please refer to the sample curve in Figure 4, as the trend reflected there is equivalent to the experimental setup of "using only global reward and removing local reward."
>
> **Q1: Clarification of the Training Process**
>
> Your understanding is perfectly correct. We currently employ an offline training approach:
> First, we use the base UMM to generate image candidates.
>
> Subsequently, we perform self-scoring on these candidates using the model's own understanding module to obtain reward values.
> Finally, the candidates and their corresponding rewards are fed into the training pipeline.
>
> We had previously considered the feasibility of an online method but did not pursue it deeply due to computational resource constraints. Systematically developing and refining an online training framework will be one of the key directions for our future work.
>
> **Q2: Local Reward Ablation**
>
> This has been answered in our response to W3.
>
> **Q3: Prompt Template Issue**
>
> As detailed in W2, we would like to add that we did not consider any adjustments for downstream validation. Our scoring instruction design is general-purpose. It is solely intended to constrain the model's output to fit the template rules, thereby increasing the success rate of programmatically extracting the reward values.
>
> **Q4: ReCA and External VLM Experiment Comparison**
>
> The relevant results are provided in the table (in W2). Our results are higher than those achieved by models trained with either external VLM scoring or the ReCA method.
>
> If you are satisfied with our response, please consider updating your score. If you need any clarification, please feel free to contact us.

---

### Author Response · Authors · 2025-11-17

Dear ACs/PCs and Reviewers,

We sincerely thank the reviewers for acknowledging the strengths of our method.

We appreciate that the reviewers recognized we have clearly identified a real and fundamental problem in Unified Multimodal Models (UMMs): the disconnect between understanding and generation capabilities. The reviewers also acknowledged our proposal of an elegant and self-sufficient self-rewarding solution to address this, which importantly requires neither external reward models nor new human-annotated data (Reviewers YiSP, 8hSG, bnfL).

The reviewers particularly recognized our innovative "global" and "local" dual-level reward system. This multi-scale feedback mechanism was noted as being well-motivated and technically sound, providing detailed and effective guidance for improving complex image generation tasks (Reviewers YiSP, bnfL, PxGQ).

Furthermore, reviewers noted that our method achieves substantial performance improvements and strong generalization ability across multiple T2I benchmarks (Reviewers YiSP, 8hSG, bnfL), and that the effectiveness of our approach is thoroughly demonstrated through comprehensive experiments (Reviewers PxGQ, 8hSG).

Related updates in the paper have been marked in **blue**.

---

### Author Response · Authors · 2025-11-23

Dear Reviewers,

Thank you for your constructive comments. As the discussion phase is nearing its end, we would like to briefly summarize how our revisions have addressed the shared concerns across three key aspects:

1.  **Superiority over External Models & Baselines:** We addressed the common concern regarding the necessity of "self-rewarding" (Reviewers YiSP, 8hSG, bnfL) by providing new comparisons with external verifiers (Qwen-VLs, UnifiedReward) and post-training baselines (ReCA). The results confirm that SRUM consistently outperforms these methods, validating that internal feedback is more effective than external dependencies for this task.
2.  **Robustness & Design Validity:** We clarified concerns regarding model collapse in deep fusion architectures (Reviewer 8hSG) and the specific roles of reward components (Reviewers YiSP, bnfL, PxGQ). Our additional ablations (e.g., w/o Local) and analysis demonstrate that our dual-reward mechanism effectively balances generation and understanding without degradation.
3.  **Efficiency & Reproducibility:** We provided the requested computational cost analysis (Reviewer YiSP) and hyperparameter details (Reviewer PxGQ), proving that SRUM introduces negligible training overhead and is highly efficient (scoring 6k samples in <4 GPU-hours).

We believe these responses and additional experiments have resolved your initial concerns. Given the limited time remaining, if you have any further questions, please let us know immediately so we can address them.

Best regards,
The Authors

---

### Note · Authors · 2025-12-08

I have read and agree with the venue's withdrawal policy on behalf of myself and my co-authors.